# Edge-Driven Multi-Agent Reinforcement Learning: A Novel Approach to Ultrasound Breast Tumor Segmentation

**DOI:** 10.3390/diagnostics13243611

**Published:** 2023-12-06

**Authors:** Nalan Karunanayake, Samart Moodleah, Stanislav S. Makhanov

**Affiliations:** 1Sirindhorn International Institute of Technology, Thammasat University, Pathum Thani 12120, Thailand; d6222300037@g.siit.tu.ac.th; 2King Mongkut’s Institute of Technology Ladkrabang, Bangkok 10520, Thailand; samart@it.kmitl.ac.th

**Keywords:** deep reinforcement learning, deep neural networks, breast ultrasound segmentation, Gestalt laws

## Abstract

A segmentation model of the ultrasound (US) images of breast tumors based on virtual agents trained using reinforcement learning (RL) is proposed. The agents, living in the edge map, are able to avoid false boundaries, connect broken parts, and finally, accurately delineate the contour of the tumor. The agents move similarly to robots navigating in the unknown environment with the goal of maximizing the rewards. The individual agent does not know the goal of the entire population. However, since the robots communicate, the model is able to understand the global information and fit the irregular boundaries of complicated objects. Combining the RL with a neural network makes it possible to automatically learn and select the local features. In particular, the agents handle the edge leaks and artifacts typical for the US images. The proposed model outperforms 13 state-of-the-art algorithms, including selected deep learning models and their modifications.

## 1. Introduction

Worldwide, breast cancer is the most common cancer in women [1]. If diagnosed early, the survival rates are significantly improved. In Asia, US examination is included in routine cancer screening due to the high prevalence of females with dense breast tissue [2]. Accurate segmentation assists in evaluating the type and stage of the tumor as well as its possible progression. However, US images are often hard to interpret. They require expertise and experience. Detection and segmentation of breast tumors is a difficult task even for skilled clinicians. They also present a challenge for CAD (computer-assisted diagnostic) systems. The US machine receives an echo of the emitted sound waves. The image of this echo (the US image) is characterized by speckle noise, irregular tumor shapes, uneven textures, poorly defined boundaries, poor contrast, and multiple acoustic shadows. The recent advances in convolutional neural networks (CNNs) and deep learning (DL) tools show much promise for the segmentation of the US images. Nevertheless, the learned features are often not interpretable. The training requires a large amount of ground truth data (usually thousands of images). Moreover, the DL trained on a specific set often fails when a different US machine or a different protocol is used [3]. Zhang et al. [4] notes that the “application of these DL models in clinically realistic environments can result in poor generalization and decreased accuracy, mainly due to the domain shift across different hospitals, scanner vendors, imaging protocols, and patient populations…the error rate of a DL was 5.5% on images from the same vendor, but increased to 46.6% for images from another vendor”. The nature of CNNs is to capture the local contextual information. However, they often miss tumors or produce over or under-segmented boundaries. One of the reasons is that the global structure of the object is not captured.

The proposed model is motivated by the autonomous vehicles of Braitenberg’s [5] and Reynolds’ bird flocks [6] proposed in the 1980s. Baitenberg’s seminal work in 1984 used reactive robots or vehicles with relatively simple sensor–motor connections aimed to simulate neuro-mechanisms in biological brains. In this paper, Braitenberg’s work is considered a metaphor for artificial life, suggesting that the complex behavior may be a result of a relatively simple design. The ideas are modified here. The robots do not know their final goal. They follow the rewards defined by a certain reward system. Additionally, they can be corrected by the neural network (NN), which observes the population and knows their final destination. Therefore, the NN is a metaphor for a “Universal God”, “Galactic Force”, etc. The idea is supported by artificial life (Alife) [7], based on tracing agents [8] and fusion images. It has been shown that, under certain conditions, Alife outperforms the state-of-the-art. Furthermore, Alife requires smaller training sets. The drawback of [7] is the requirement of the additional set of elasticity images, which are not always available. Further, the above ALife is based on fixed rules and requires extensive training performed manually or by a genetic algorithm. Therefore, an extension of this model is offered. The new model does not require the elasticity images and achieves the same or better accuracy. It uses a relatively small training set, and is capable of handling previously unseen data.

On this level, the basic ideas are to: (1) generate a preliminary binary mask and offset trajectories to guide the agents, (2) train the agents using the convolution network, and (3) include the RL to replace the requirement of the image fusion. The model combines multi-agent reinforcement learning, neighbor message passing, and deep learning to generate trajectories of the agents, approximating the boundary of the lesion. Message passing is based on principal neighborhood aggregation [9]. Multi-agent deep reinforcement learning (MADRL) employs the Gestalt laws (GL) [10] of shape perception to construct the reward system.

Summarizing the new elements above, the novelty of the proposed model is as follows.

-A new hybrid of the DL, Alife, and RL, has been offered. The method combines the strengths of the DL neural networks with the efficiency and simplicity of ALife and the agents’ communication and learning by RL.-A reward system based on the GL.-An original classification of the images based on the properties of the edge maps included in the training algorithm.-Verification by previously unseen data.

We also consider it valuable that the proposed bio-inspired model does not use the standard genetic framework which often slows down the optimization. The agents are identical, they do not evolve, do not mate, and do not have leaders. This makes it possible to achieve fast computational times, e.g., 5–20 s per image 500 × 500 (Section 7). The agents communicate and try to maximize their rewards. Hence, the model is able to generate complicated patterns, fit irregular structures, and segment the image efficiently. The proposed MADRL algorithm has been tested on two US datasets against 13 state-of-the-art algorithms. They include active contours (AC), edge linking, level set methods (LS), superpixels, machine learning, deep learning algorithms, and their modifications. The numerical results reveal that the MADRL outperforms its competitors in terms of accuracy when applied to high-complexity images. Note that the paper performs cross-field testing, i.e., compares the segmentation methods based on different principles. The most popular and original algorithms from six classes have been selected. First of all, consider the deformable models, i.e., AC and LS. They have been massively used for edge-based segmentation and extensively used for segmentation of the US images of the breast and thyroids. The latest survey [11] considers the deformable models of the most frequently used approach for segmentation of the US images. The  adaptive diffusion flow [12] is one of the most remarkable versions of the AC. It establishes a joint framework between the classical gradient vector flow and the image restoration. The recent AC energy-based pressure approach [13] introduces an “energy balance” between the inner region and the outer region. Finally, the hybrid AC-LS method [14] based on the local variations of the grey level shows excellent results on various synthetic and real medical images. According to a survey in [7], these three algorithms outperform 29 top models from this class (https://sites.google.com/view/edge-driven-marl-siit-biomed/additional-references, accessed on 28 June 2023). The pioneering LS work is Osher, 1988 [15]. The propagating fronts, with velocity depending on the front curvature, became revolutionary in the image processing of the 2000s. The distance-regularized edge-based DRLSE [16] is one of the most popular. Introducing the third dimension solves the problem of the multiple ACs and their possible merge. We refer the reader to various modifications of the LS given in the excellent review [17]. The DRLSE, the saliency-driven region/edge-based LS [18] and a correntropy-based LS [19] have been shown to outperform eight top algorithms from this class (https://sites.google.com/view/edge-driven-marl-siit-biomed/additional-references, accessed on 28 June 2023).

Edge linking methods connect pieces of the object boundary of the edge map into a closed contour subject. The algorithm [20] is based on edge tracing and a Bayesian contour extrapolation. The ratio contour method [21] encodes the GL laws of proximity and continuity combined with a saliency measure based on the relative gap length and average curvature. Some of these ideas have been used in our model. The above models outperform four edge-linking procedures (https://sites.google.com/view/edge-driven-marl-siit-biomed/additional-references, accessed on 28 June 2023). Note that the early variants apply the graph-based approach, where the graph represents candidate parts of the boundary, and the weights represent the affinity between them. The graph is partitioned to optimize a selected utility function. Further, the edge linking evolved in different directions. Some prominent examples are connected components [22], pixel following [23], and the dot completion game [24].

A recent review [25] claims that the superpixel models are one of the most important tools for image segmentation. In particular, the simple linear iterative clustering (SLIC) proposed in [26] is one of the most efficient. We compare the RL with the recent version of SLIC [3], which integrates the NN and k-NN techniques. Our second superpixel reference is [27]. This model applies the superpixels to training, whereas segmentation is performed by the LS. The prior knowledge regarding the possible shape of the object is included in the LS functional.

Machine learning (ML) (deep learning and neural networks) is one of the most promising fields of study aimed at processing and recognizing US images [28,29]. The ML algorithm adapts itself to generate the required features instead of selecting them manually. The most successful models are CNNs and DL networks [30,31].

We consider U-Net [32] as one of the most prominent and influential models of this type. The U-net is often treated as “the benchmarking DL model for medical image processing” [33]. A hybrid of the ML and the Chan–Vese algorithm [34] is proposed by [35]. It combines the k-NN method and the support vector machine. We also test against a semi-supervised DL generative adversarial network proposed by [36]. The DL applies the multi-scale framework and combines the ground truth and the probability maps obtained from the un-annotated data. The model outperforms the ASDNet, ZengNet, and DANNet (https://sites.google.com/view/edge-driven-marl-siit-biomed/additional-references, accessed on 28 June 2023). We also test against the selective kernel [37]. Currently, the selective kernel is one of the most developed U-Net algorithms, which outperforms many recent modifications of the U-Net. It should be noted that the above is not a complete review of the state-of-art of medical image segmentation. The above models and some of their modifications have been selected as the benchmark for testing. The selection criteria are publications in a reputable journal, references, tests against similar and cross-field models, and availability of the code. Our numerical results show that the proposed DL reinforcement learning outperforms 13 selected models from different classes (cross-field testing). The advantage against the second-best performing model on the images characterized by a high complexity is as high as a 97% (success rate) against 77%. The results are subjected to the selected set of images and the proposed accuracy measures. The developed code is applicable to breast cancer diagnostics (automated breast ultrasound),  US-guided biopsies, as well as for projects related to automatic breast scanners. A demo video illustrating the algorithm is at (https://drive.google.com/file/d/1kqW68mdQ1QmkasA1gnXNFRRaavhvpA-M/view?usp=drive_link, accessed on 30 August 2023).

## 2. Reinforcement Learning

Reinforcement learning (RL) is a strategic framework in which the agent learns to perform a specific task through a series of actions and rewards.

### 2.1. Concept of Reinforcement Learning

The concept of RL has its roots in psychology and biology. It was introduced in the early 20th century by animal learning theorists, such as Pavlov and Thorndike. Further, Sutton and Barto [38] used these ideas for ML and Alife. The multiple agent RL models introduce cooperating agents to handle complex tasks [39]. As the DL evolved, the RL was combined with DL to become deep reinforcement learning (DRL) [40], where DL acts as the control center. Nowadays, the DRL is becoming increasingly popular to further explore “the rabbit hole of the interactive AI” [39,41].

The recent examples are the DRL in robotics [42,43], simulations [40], language processing [44], autonomous vehicles [45], computer vision [41,46], and medical image processing [47]. Some popular DRL variants are deep Q-networks [48], deep deterministic policy gradient [49], trust region policy optimization [50], and proximal policy optimization (PPO) [51]. As opposed to supervised learning, DRL offers sequential decision-making [41], which relies on exploratory experiences. This is particularly suitable for image segmentation. The RL models are able to develop a strategic path toward an optimal collective solution beyond local temporal gains. In this sense, RL tries to mirror human learning mechanisms to adapt to variable, complex environments.

### 2.2. Reinforcement Learning for Medical Image Segmentation

Segmentation in medical imaging is usually performed by supervised learning. This requires extensive annotated data, which is often lacking in the field. The RL partly circumvents these limitations since the RL model can be trained on smaller sets. During the last two decades, the RL has not been particularly popular in medical image processing. The main application of this algorithm was the Q-learning for obstacle avoidance in robotics [52]. However, the recent review [53] considers image processing to be one of the key applications of Q-learning. An early attempt to adapt Q-learning to detect the segmentation threshold [54], evaluates the action probability using a variant of the Boltzman policy [55]. An experienced operator guides the agents via the graphics user interface. References [56,57] propose an online RL evaluation. The US image is divided into sub-areas, and the actions are defined as adjusting the thresholds and the structural elements. The agents are rewarded by using the groundtruth data. An RL segmentation of the computed tomography images is proposed in [58]. The author proves that the approach requires a smaller set of training data compared to other methods. The Q-matrix is used to fix the segmentation thresholds. The approach is fully automated and parallelized. However, the above models show a lack of adaptation and generalization. In particular, the algorithm often fails when applied to unseen data. To circumvent these problems, ref. [59] proposes an online RL, where the agents memorize their previous interactions. Segmentation is performed by using the instantiating initial points and the user feedback to update the state–action space. The model has been applied to the segmentation of cardiac MRIs. A multi-stage DRL combined with the actor-critic algorithm has been proposed in [60]. The value and policy networks are variants of ResNet-18 [61]. The model has been applied to MRIs and retinal images.

A recent model [62] performs segmentation of the MRI images for the diagnosis of cardio-diseases. The method combines double Q-learning with the deep neural network [63] trained to find the true edge. The  DRL has also been proposed for parameter optimization in [64] and initialization of the object mask in [65]. Image segmentation and classification are interconnected problems often performed simultaneously. The examples of such models are [66,67,68]. The DRL for image registration is used by [69,70]. According to the recent study [47], the advantages of DRL over the conventional schemes are the reduction of training data, the  reduced memory, and the ability to discover new features during the sequential search for the optimal solution.

## 3. Agents

The agents are trained by the RL using the edge map, the offsets, and the grayscale. The successful trajectories connecting broken edges constitute a set of candidates of the boundary. The agents communicate and adapt their movement to maximize the reward. The boundary constitutes a small fraction of the edge map (about 1–5%). In order to reduce the search space, the model creates an attention mask. Offset trajectories are generated within the mask to guide the agents. The mask is generated using histogram equalization, a variant of the superpixel decomposition [3] combined with neutrosophic clustering [71]. Figure 1 illustrates generation of the mask.

### 3.1. Offsets

The offsets are generated by the trim-and-join algorithm [72] and are parameterized by the cubic splines. For practical purposes, we experimented with the internal and external masks to further optimize the approach. However, the generation of the internal mask has not been fully automated. Figure 2 visualizes offset generation.

### 3.2. Message Aggregated Deep Reinforcement Learning

The proposed message aggregated/multiple agent DRL algorithm is based on the Gestalt Laws of shape perception. It should be noted that the GL has been included in many image processing and pattern recognition algorithms. We refer the reader to the excellent review [73] and the recent publications [74,75]. The proposed MADRL includes:The convolution encoder (feature extraction);Message passing and feature aggregation unit;The (deep neural network) DNN integrated with the DRL.

The DNN generates the stochastic policy π. Based on the observation o(t), the action of the agent is a step in a certain direction (the velocity vector $v_{i}\left(t\right)$). The reward function is a linear combination of multiple rewards (Section 3.4). A gradient-based RL is used, where the parameterized policies are optimized according to their expected returns.

### 3.3. Observation Space

The agents follow the offsets in either the clockwise (CW) or counterclockwise (CCW) direction. During their lifetime, they observe the environment within a specified window $w_{i}$. Their actions are generated by the partially observable Markov decision model proposed in [76,77]. The model includes the state space, the action space, the transition function, the reward function, the observation space, and the observation probability distribution. The agents are identical. The agent *i* reads the observations $o_{i}\left(t\right)$ and aggregates features $f_{i}\left(t\right)$ from the neighboring agents. Based on the above, the agent evaluates the rewards $r_{i}\left(t\right)$ and selects the action $a_{i}\left(t\right)$. The observation vector $o_{i}\left(t\right)$ is characterized by:The state: the agent is tracing the edge (state = edge) or flies outside the edge (state = free).The offset velocity is $v_{i}^{s}\left(t\right)$. If the agent is free, $v_{i}^{s}\left(t\right)$ is the tangent to the closest point on the offset trajectory.The average curvature of the trajectory along an interval $\left[t-t_{2},t\right]$ is evaluated by

(1)$$s_{i,C}\left(t\right)=\frac{1}{t_{2}}\int_{t-t_{2}}^{t}\left|\kappa_{i}\left(s\right)\right|ds,$$ where $\kappa_{i}\left(s\right)$ is the curvature, and *s* is the parametric variable along the trajectory.

Further, the agent has a memory, which includes the last three frames $M_{i}^{\prime }\left(t\right)=(M_{i}\left(t-2\right)$, $M_{i}\left(t-1\right),M_{i}\left(t\right))$. Hence, the observation space includes the observation vector, the aggregated feature vector, and the grid maps $M_{i}^{\prime }$. Information exchange allows the generation of complicated patterns to fit the unknown boundary [39,78]. The neighborhood aggregation [9] employs the mean, maximum, minimum, and standard deviation. The features undergo amplification and attenuation and are saved in the aggregated feature vector $f_{i}\left(t\right)$. The key elements of the system are shown in Figure 3.

Finally, note that the feature aggregation captures the behavior of the agents and their interactions with the environment, whereas the convolution unit processes stationary pixel data.

### 3.4. Reward Function

The reward function based on the GL is considered as the domain knowledge [79]. The signals are dynamic. They control the perception of the boundary shape and the contour linking [80,81]. Recall that the GL includes continuity, closure, and proximity. Therefore, the reward $r_{i}\left(t\right)$ is given by,
(2)$$r_{i}\left(t\right)=w_{c,1}r_{i}^{continuity,1}\left(t\right)+w_{c,2}r_{i}^{continuity,2}\left(t\right)+w_{cl}r_{i}^{closure}\left(t\right)+w_{p}r_{i}^{proximity}\left(t\right)+w_{d}r_{i}^{density}\left(t\right),$$
where $w_{a}\;a=c,cl,p,d$ are the corresponding weights. The continuity reward is given by
(3)$$r_{i}^{continuity,1}\left(t\right)+=\left\{\begin{array}{cc} \Delta r & \mathrm{if}\;\;s_{i,C}\left(t\right)\leq s^{\prime },\\ 0 & \mathrm{otherwise}, \end{array}\right.$$
where $s_{i,C}\left(t\right)$ is defined by Equation (Equation 1). Recall that the agents are following the offsets in the CW and CCW directions. When the CW and CCW agents collide they connect the corresponding edges. Hence, we consider the angle $\theta_{i,k}$ between the trajectory *i* and *k* at the collision point. Additionally, the agents can not live long outside the edge. If  $t_{\mathrm{free}}>t_{live}$, the agent dies. Therefore, the continuity reward is incremented as follows
(4)$$r_{i}^{continuity,2}\left(t\right)+=\left\{\begin{array}{cc} \Delta r & \mathrm{if}\;\;\theta_{\min}\leq \theta_{i,k}\leq 180{}^{\circ},\\ 0 & \mathrm{otherwise}. \end{array}\right.$$

The proximity is rewarded as follows
(5)$$r_{i}^{proximity}\left(t\right)+=\left\{\begin{array}{cc} \Delta r,\;\; & \mathrm{if}\;\;t_{free}\leq t_{cont}\\ 0 & \mathrm{otherwise}. \end{array}\right.$$

If the agent returns to a neighborhood of its departure, the reward is incremented
(6)$$r_{i}^{closure}\left(t\right)+=\Delta r.$$

The density reward is given by
(7)$$r_{i}^{density}\left(t\right)+=\left\{\begin{array}{cc} \Delta r & \mathrm{if}\;\;d_{i}>d^{\prime }\\ 0 & \mathrm{otherwise}. \end{array}\right.$$

The notations above are self explanatory. The weights are a part of the training set (Section 4 and Section 7).

### 3.5. Network Architecture

The proposed network is shown in Figure 4. The grid maps, observations, and rewards are the input of the DNN.

The DNN consists of six hidden layers (three convolution and three dense layers). The grid maps $M_{i}^{\prime }$ are used to extract the low-level features of the agent’s environment. The resulting features are then combined into the feature vector. The first convolution layer applies 32 two-dimensional filters with a kernel size of three and a stride of one. A max-pooling layer with kernel two and stride two follows. The second and third convolution layers consist of 64 and 128 2D filters, respectively, with kernel three and stride one. They are followed by max-pooling layers with the kernel two and stride two. The feature maps are flattened into a 256-dimensional vector. The flattening layer concatenates $o_{i}\left(t\right)$, $f_{i}\left(t\right)$, and $r_{i}\left(t\right)$ and feeds the next two fully connected layers with 128 rectifier units. The final layer is a fully connected layer with a sigmoid activation [82].

## 4. Network Training

The PPO [51] is used for training by an on-policy gradient algorithm [60] based on the DRL actor–critic techniques. The PPO produces smooth gradient updates to ensure a stable policy space and reduced hyperparameter tuning. The agent trajectories are standardized and used to construct the surrogate loss $L_{\theta }$ and the least square error ($L_{2}$ loss) $L_{\phi }$. The loss function is minimized using the Adam optimizer [83]. The policy is updated over $E_{\pi }$ epochs and the value function is updated over $E_{\phi }$ epochs.
(8)$$L_{\theta }={\hat{E}}_{t}\left[\min\left(r_{t}\left(\theta \right){\hat{A}}_{t},\mathrm{clip}\left(r_{t}\left(\theta \right),1-\epsilon ,1+\epsilon \right){\hat{A}}_{t}\right)\right],$$
(9)$$r_{t}\left(\theta \right)=\frac{\Pi_{\theta }\left(a_{i}\left(t\right)|o_{i}\left(t\right)\right)}{\Pi_{old}\left(a_{i}\left(t\right)|o_{i}\left(t\right)\right)},$$
where ${\hat{E}}_{t}$ denotes a stochastic estimate of the expected value over a mini-batch of transitions. The clipping parameter ϵ controls the range of the probability ratio $r_{t}\left(\theta \right)$ used for the update. ${\hat{A}}_{i}\left(t\right)$ is a generalized advantage estimate [84] that smooths the discounted rewards and reduces the variance of the policy gradients to make the training stable. In addition, it indicates how good a particular state is. The estimate is defined by
(10)$${\hat{A}}_{i}\left(t\right)=\delta_{i}\left(t\right)+\left(\gamma \lambda \right)\delta_{i}\left(t+1\right)+\ldots +{\left(\lambda \gamma \right)}^{T_{i}-t+1}\delta_{i}\left(T_{i}-1\right),$$
where λ is the smoothing factor, γ is the discount factor for the future rewards, $T_{i}$ is the time step of the episode, and *i* is the index of the agent. The term $\delta_{i}$ implies the advantage of the new state over the previous state. It takes into account the immediate reward $r_{i}\left(t\right)$ and the expected future value $V_{\pi }s_{i}\left(t+1\right)$, discounted by a factor of γ.
(11)$$\delta_{i}\left(t\right)=r_{i}\left(t\right)+\gamma V_{\pi }s_{i}\left(t+1\right)-V_{\pi }s_{i}\left(t\right).$$

The value function $V_{\pi }s_{i}$ is used to update the policy function. The $L_{2}$ loss is given by
(12)$$L_{\phi }=\frac{1}{K}\sum_{i=1}^{K}\sum_{t=1}^{T_{i}}{\left(V_{\phi }\left(s_{i}\left(t\right)\right)-{\hat{V}}_{i}\left(t\right)\right)}^{2},$$
where *K* is the total number of trajectories, $V_{\phi }\left(s\right)$ is the value function approximated by the network by the parameters ϕ. ${\hat{V}}_{i}\left(t\right)$ is the estimated discounted return.

Further,
(13)$${\hat{V}}_{i}\left(t\right)=V\phi \left(s_{i}\left(t\right)\right)+\sum {t^{\prime }=t}^{T_{i}-1}\gamma^{t^{\prime }-t}r_{i}\left(t^{\prime }\right).$$

Note that the policy and value networks are characterized by the same architecture but their parameters are not shared. The optimal policy $\pi_{\theta }$ is being generated by the networks simultaneously.

The pseudocode of the training algorithm is presented as Algorithm 1, while Table 1 lists its hyperparameters.
**Algorithm 1** Multi-agent deep reinforcement learning  1:**for** iterations =1,2,3… **do**  2:    Initialize θ,ϕ  3:    **for** Agent i=1,2,3,…,N **do**  4:        // data collection  5:        Run the policy $\pi_{\theta }$ for $T_{i}$ time steps  6:        Collect observations, rewards and actions $o_{i}\left(t\right),r_{i}\left(t\right),a_{i}\left(t\right)$, where $t\in [0,T_{i}]$  7:        Estimate the advantage ${\hat{A}}_{i}\left(t\right)$  8:        Break $if\sum_{i=1}^{N}T_{i}>T_{max}$  9:    **end for**10:    //update the policy function11:    **for** $j=1,2,3,\ldots ,E_{\pi }$ **do**12:        Calculate surrogate loss $L_{\theta }$13:        Update θ with the learning rate $lr_{\theta }$ by the Adam optimizer based on $L_{\theta }$14:    **end for**15:    //update the value function16:    **for** $k=1,2,3,\ldots ,E_{\phi }$ **do**17:        Calculate the surrogate loss $L_{\phi }$18:        Update ϕ with the learning rate $lr_{\phi }$ by the Adam optimizer based on $L_{\phi }$19:    **end for**20:**end for**

### Training Data

The network has been pre-trained using the transfer learning scheme [85]. The MADRL has been tested on 1000 US images from http://www.onlinemedicalimages.com (accessed on 17 June 2023) of Thammasat University Hospital, Thailand, Bangkok (Philips iU22 US machine) and https://www.ultrasoundcases.info (accessed on 20 July 2023) from The Gelderse Vallei Hospital, Ede, The Netherlands (Hitachi US machine). Three experienced radiologists with Thammasat University Hospital provided the ground truth using the electronic pen on a Microsoft Surface Tablet. The final ground truth is obtained by the majority rule. The image resolution ranges from 200×200 to 600×480 with a 60:40 ratio for training and evaluation. The training edge maps are categorized according to their complexity. The categories include the complexity of the shape, the length of the gaps relative to the contour, and the maximum gap on the contour (Section 5). The approach allows us to find a suitable solution in a less difficult environment fast and to progressively advance the complex environments. When stability and convergence are established in each category, the training is completed.

## 5. Experimental Results

We introduce the following quality measures.

### 5.1. Contour Based Metrics

The Hausdorff distance is defined by
(14)$${\mathrm{dist}}_{H_{1}}\left(X,Y\right)=\max(\max_{a\in X}\min_{b\in Y}∥a-b∥,\max_{b\in Y}\min_{a\in X}∥a-b∥),$$
where *X* is the ground truth contour, and *Y* is the resulting contour.

The average Hausdorff distance is defined by
(15)$${\mathrm{dist}}_{H_{2}}\left(X,Y\right)=\max(\frac{1}{L_{X}}\sum_{a\in X}\min_{b\in Y}∥a-b∥,\frac{1}{L_{Y}}\sum_{b\in Y}\min_{a\in X}∥a-b∥),$$

$L_{X}$ and $L_{Y}$ are the lengths of the resultant contours of two contours *X* and *Y*, respectively.

The relative Hausdorff distance is defined by
(16)$${\mathrm{dist}}_{H_{3}}\left(X,Y\right)=\frac{1}{L_{X}}{\mathrm{dist}}_{H_{2}}\left(X,Y\right)\;100\%.$$

Note that ${\mathrm{dist}}_{H_{1}}$ and ${\mathrm{dist}}_{H_{2}}$ evaluate the absolute difference between the contours. However, ${\mathrm{dist}}_{H_{3}}$ is normalized to the length of the contour. Therefore, if the set includes objects of a different size (which is the case) ${\mathrm{dist}}_{H_{3}}$ is preferable.

### 5.2. Area-Based Metrics


(17)Recall:Rec=TPTP+FN100%,(18)Precision:Pre=TPTP+FP100%,(19)Accuracy:Acc=TP+TNTP+TN+FP+FN100%,(20)JaccardIndex:Jac=TPTP+FP+FN100%,(21)DiceIndex:Dice=2TP2TP+FP+FN100%.


Here, TP, FP, TN, and FN denote true positives, false positives, true negatives, and false negatives. Note that Acc is proportional to *TN*. That is why it is often large even when the segmentation quality is poor. The Dice and the Jaccard indices are often considered the most reliable area-based measures in medical image processing [86].

### 5.3. Successful Segmentation

It is often the case that the model produces a few outliers with a low accuracy. In this case, the average accuracy does not clearly represent the performance of the method. Therefore, we collect the outliers into a separate group and introduce the accurate segmentation ratio $SG_{\mathrm{ratio}}$ defined as the percent of the successful results relative to the entire number of trials. The segmentation is considered successful if $H_{3}\leq 2\%$. Only the successful cases are used to calculate the evaluation metrics. In order to show the actual accuracy, the results include a $SG_{\mathrm{ratio}}$.

### 5.4. Image Categorization

Conventional categories of the US images (Figure 5) usually are not suitable for edge-map-based models.

Therefore, in addition to the above, we categorize the edge maps. The proposed categories allow us to train and assess the performance of the algorithm and identify the directions of improvement. Note that characterizing the complexity of the US images by signal-to-noise ratio (SNR), contrast-to-noise ratio (CNR), or similar measures is often pointless. Many US artifacts are not the noise. They represent a real ultrasonic echo of an actual human tissue rather than a noise produced by the equipment or the environment.

#### Image Complexity

We define (1) the ratio of the total length of the gaps to the length object boundary $L_{g}$, (2) the ratio of the max gap to the length of the boundary $L_{g,\max}$. The complexity *C* relative to $L_{g}$ is defined using the standard deviation as follows. If $L_{g}<L^{\prime }$, then complexity C = B (baseline), otherwise C = T (tough). Here, $L_{g}^{\prime }={\tilde{L}}_{g}+\sigma_{g}$, where ${\tilde{L}}_{g}$ represents the mean and $\sigma_{g}$ the standard deviation. The complexity relative to the maximum boundary gap $S_{\max}$ is defined similarly. The complexity of the shape *S* is defined by the difference between the area of the tumor and the area of the smallest embedding circle. Based on the above definitions, the complexity of the US images is encoded by $S|C_{\max}|C$, where S,C and $C_{\max}$ is either *B*-easy or *T*-difficult. For example, B|B|B stands for the simplest images, while T|T|T stands for the most complex cases (Figure 6).

## 6. Numerical Experiments

MADRL has been tested against 13 state-of-the-art methods discussed in the introduction. We include the active contours: ADF-AC [12], ALF-AC [14], EBF-AC [13], level set methods: DR-LS [16], CB-LS [19], SR-LS [18], edge linking: BS-EL [20], CR-EL [21], superpixels: PK-SP [27], SC-SP [3], and the machine learning/deep learning algorithms ML-AC [35], SS-GAN [36], and S-U-NET [37]. Figure 7 shows samples of the segmented US images by MADRL vs. the benchmark methods for different levels of complexity. The Figure is complemented by Table 2, Table 3, Table 4, Table 5, Table 6, Table 7, Table 8 and Table 9 corresponding to the increasing complexity. Table 2 shows the segmentation results for the simplest complexity B|B|B. Nine methods produce $SG_{ratio}=100\%$. However, $SG_{ratio,\mathrm{BS}\text{-}\mathrm{EL}}$ and $SG_{ratio,R\text{-}\mathrm{EL}}$ dropped by 25% and 16%. The methods with 100% segmentation show the maximum $H_{3,\mathrm{EBF}\text{-}\mathrm{AC}}=1.18\%$ (acceptable). All methods manage Dice≥80, where $Dice_{\mathrm{BS}\text{-}\mathrm{EL}}=80$ is the lowest. The best $Dice_{\mathrm{MADRL}}=96.$

Consider the impact of $L_{g}$ and $L_{g,\max}$ on the efficiency of the US segmentation. Table 3 and Table 4 show the results for B|T|B and B|B|T. Clearly, the maximum size of the gap $L_{g,\max}$ has a greater impact on accuracy than their total length. In particular, eight methods for B|B|T have $SG_{ratio}=100\%$ and three methods fall below 90%. However, in the case of B|T|B only six methods remain in the 100% category (including the proposed MADRL). Moreover, the number of successfully segmented images drops below 90% for the five methods. The lowest $Dice_{\mathrm{BS}\text{-}\mathrm{EL}}=65.38$ with $H_{3,\mathrm{BS}\text{-}\mathrm{EL}}=2.6$.

The results for the complexity B|T|T are given by Table 5. It combines a large total gap length, a large maximum gap, and a simple shape S=B. Only SR-LS, PK-SP, SC-SP, SS-GAN, S-U-NET, and the proposed MADRL achieve the segmentation rate 100%. Nonetheless, the RL has only a marginal improvement over the S-U-NET. For instance, $DICE_{S\text{-}U\text{-}\mathrm{NET}}=92.02$ and a $H_{3,S\text{-}U\text{-}\mathrm{NET}}=0.74\%$, while the $Dice_{\mathrm{MADRL}}=93.05$ and a $H_{3,\mathrm{MADRL}}=0.53$.

Table 6 and Table 7 shows the significance of the tumor shape. Even T|B|B presents a challenge. The occurrence of a leakage at the boundaries and irregularity of the geometry of the tumor reduce the success rate. The exception is S-U-NET and the proposed MADRL. Note that SR-LS, PK-SP, SS-GAN, and S-U-NET maintain an acceptable success rate above 90%. However, the proposed method outperforms them with SG=100% the best $Dice_{\mathrm{MADRL}}=91$ and the best $H_{3,\mathrm{MADRL}}=0.61$.

The complexity T|T|B indicates an irregular shape and a considerable edge leakage (Table 8). This configuration finally breaks S-U-NET although its success rate remains above 90%. The ACs and LSMs fail. SR-LS achieves a modest $SG_{\mathrm{ratio}}=73\%$. The edge-linking methods BS-EL and CR-EL fail with a success rate below 30%. However, MADRL stands out with $SG_{\mathrm{ratio},\mathrm{MADRL}}=100\%$, Dice=89.6 and $H_{3}=0.84$.

The most difficult type, T|T|T (Table 9), shows a low success rate for the majority of competing methods. The edge linking methods BS-EL and CR-EL have an unacceptable $SC_{ratio}=0$. The complexity of the shape combined with the edge leaks leads to the failure of the AC and LS models. The minimum and maximum success rates of the ACs are 13% and 27%, respectively. The LS is slightly better. The minimal success rate $SG_{ratio,\mathrm{DR}\text{-}\mathrm{LS}}=36\%$ and the maximum $SG_{ratio,\mathrm{SR}\text{-}\mathrm{LS}}=63\%$. One of the main disadvantages of the deformable shapes is a strong dependence on the initialization and the inability to handle strong edge leaks. The T|T|T images are irregular, having acoustic shadows and artifacts. As the result, the edge detector generates strong false boundaries and the edge-based AC and LS produce a poor segmentation.

Further, the drawback of the superpixel methods is a potential loss of the true edge if it gets inside a generated superpixel. SC-SP and PK-SP show a success rate of less than 60%. Their Dice<80. Our main competitor S-U-NET has $SG_{ratio}=77\%$. We conjecture that this is due to the complexity of the shape, insufficient training data, and the inability of the DL methods to capture the global information. This comment applies to all ML and DL methods presented in the tables. Note that only the S-U-NET achieves a reasonable Dice=79, which is slightly below the threshold of 80. Further, $H_{3,S\text{-}U\text{-}\mathrm{NET}}=1.44\%$ is also an acceptable result. However, MADRL has a significantly higher SG=98%, Dice=90, and $H_{3}=0.83$.

One of the main reasons for the failure of the DL methods is that the algorithms do not understand the global context. As opposed to that, MADRL combines the DL which allows for the automatic feature extraction with self-trained reinforcement agents capable of communicating the information throughout the entire population. Another reason is a lack of annotated data. Usually, a pure DL model requires one-tenth of thousands of samples. However, the proposed model employs only about 2000 annotated images.

## 7. Scalability and Parameters of the Agent Locomotion

Scalability is the measure of the decrease in performance of the model in response to changes in the scale of the input data. The numerical experiments show that the computational time is proportional to the length of the boundary of the tumor. Figure 8 shows the average computational time for the tumor length ranging from 500 to 5000 pixels. Roughly, this corresponds to the images ranging from 100 × 100 to 1000 × 1000 pixels. Fitting the data to the interpolating polynomial produces the best fit for the linear function. The coefficients at $L_{b}^{2}$ and $L_{b}^{3}$ are less than $10^{-7}$. Therefore, $t=O(L_{b})$. The relationship between the number of agents, the length of the boundary, and the complexity of the tumor (eight categories) is an open problem that requires further research and massive numerical experiments.

The optimal values of the reward weights introduced in Section 3.4 obtained by training are given in Table 10 below. The table shows that the continuity measured by the average curvature (continuity 1) and the angle of collision of the CW and CCW agents (continuity 2) constitute the most important rewards. The proximity is important as well. However, if the boundary gap is large but two pieces of the boundary fit into a smooth curve, it is rewarded accordingly.

The model is implemented in Python 3.8 in OS Linux with the standard Intel Core i7-10700 processor with 16 GB RAM and NVIDIA GTX 1080 graphics.

## 8. Segmentation Ratio and Standard Deviation

The heatmap in Figure 9 shows the segmentation ratio of MADRL compared to the best eight competing methods, namely, S-U-NET, SS-GAN, SR-LS, PK-SP, SC-SP, CB-LS, ML-AC, and DR-LS. All models have a score of 100% at the simplest levels B|B|B and B|B|T. At the level B|T|B, every model except ML-AC and DR-LS maintains 100%. However, as the complexity increases from B|T|T to T|T|T, the models display a significant decline. However, MADRL stands out as the most robust model, upholding 100% until T|T|B (level 7) and achieving 97.6% even at the most challenging level T|T|T. While the DL models S-U-NET and SS-GAN come close to MADRL, none of the models match MADRL’s consistently high performance. In contrast, DR-LS shows the fastest decline, with scores dropping to 77.78 at level B|T|T, plummeting to 36.36 at T|T|T. Consequently, while all other models’ performance tends to decline with increasing complexity, MADRL remains consistent across the eight levels. Figure 10 presents the Dice index and its standard deviation (SD) for the best eight segmentation models at the complexity levels. MADRL demonstrates stable performance, as indicated by the lowest SDs. The *p*-values of the difference between the second best method and MADRL show an excellent statistical significance (*p* < 0.001) for Dice and $H_{3}$ in the last four categories.

## 9. Conclusions and Future Work

A new multi-agent reinforcement learning (MADRL) algorithm tailored for breast US segmentation has been proposed and analyzed. The combination of the DL with the tracing RL agents demonstrates significant improvements over the existing state-of-the-art. The method shows an excellent performance across a variety of complex US images, effectively overcoming challenges, such as tumor heterogeneity, boundary leakage, and the global shape context problem. Notably, unlike the CNNs, which learn to extract patterns and features from the data, the MADRL understands the structure of the image through the dynamics of the agents.

The proposed model is motivated by the autonomous vehicles of Braitenberg [5] aimed to simulate neuro-mechanisms in biological brains. The ideas of Braitenberg have been successfully adapted to the segmentation of US images. The new reward system based on the Gestalt Laws of shape perception has been developed and implemented. The system includes two continuity measures, proximity measure, density, and closure measure. The numerical training shows that 50% of the rewards can be attributed to the continuity and 20% to the density measure. The final solution regarding the locomotion of the agents is generated by the DNN, which observes the entire population and knows their goal. This global knowledge of the image has been achieved using the dynamics of the agents and their ability to exchange information. Another source of information is the classification of the training images into eight categories. Therefore, the system is fed by samples with increasing complexity. This allows the DNN to adapt to the image features using an iterative approach, i.e., start from the simplest category and finalize the training at the hardest level.

The numerical experiments test the proposed MADRL against 13 benchmark methods. The analysis of the results shows that the majority of the competing algorithms perform satisfactorily at the first 3–4 levels of complexity. The complexity of the tumor shape has the most profound impact on the accuracy of the results. When the shape of the tumor changes from simple to complex, 12 benchmark methods from 13 fail for some images. The success rate changes from 45% for conventional edge linking to 96% for a variant of the generative adversary network [36]. Only S-U-NET [37] retained the original 100% success rate with the Dice coefficient 88%. The success rate S-U-NET drops below 100% only when the shape of the tumor is complex and the max boundary gap is large and when all three criteria indicate high complexity. In this case, $SG_{\mathrm{ratio},S\text{-}U\text{-}\mathrm{NET}}=93\%$ and 91%, respectively. However, MADRL outperforms its main competitor, keeping $SG_{\mathrm{ratio},\mathrm{MADRL}}=100\%$ and Dice≥88%.

The reward function is a variant of the mathematical representation of the Gestalt Laws of shape perception. The numerical experiments with the reward function show that the continuity of the trajectory constitutes the most important reward (over 50%). The density (a variant of proximity) is the second important parameter (around 19%).

The model shows the lowest standard deviation and an excellent statistical significance in terms of the *p*-values.

The computational time depends on the size of the tumor. Our preliminary experiments show that the computational time is a linear function of the length of the tumor boundary. In other words, the model shows excellent computational performance. The average time required for the 500 × 500 image ranges from 5 to 20 s. The model processes an image 1000 × 1000 at 30 s.

Note the limitations of this study. While our method has shown significant progress in segmenting US tumors, the results may vary depending on the quality of the images and the complexity of the tumor when the complex images are from an unseen database. The model has many training parameters. Most likely they require substantial re-training and transfer learning if applied to different types of medical imagery, such as MRI, CT-scans, etc. However, the design of a universal segmentation model is outside the scope of this study. One of the specific features of the US images is a sufficiently smooth boundary of the tumor (although globally the shape can be very irregular). This helps the agents accurately approximate the boundary. The model requires modifications if applied to objects with many sharp turns. In this case, the agents “wiggle” at the corners. This requires modifications and specific training.

Our future research focuses on optimizing this approach for real-time segmentation and exploring its potential for classification of the breast tumors. Another goal is to apply it to US imagery of different human organs. The most probable candidates are the US images of thyroids. From the viewpoint of computer science, one of the most interesting questions is the relationship between the number of agents, the length of the boundary, and the complexity of the tumor (eight categories). This open problem requires further research and massive numerical experiments. From the viewpoint of clinical application, the speed remains one of the most important factors. Doctors do not want to wait 20 s. The result must appear immediately on the screen, i.e., 2–3 s. However, the simplicity of the model mentioned above makes it possible to conjecture that this goal is achievable.

## Figures and Tables

**Figure 1 diagnostics-13-03611-f001:**
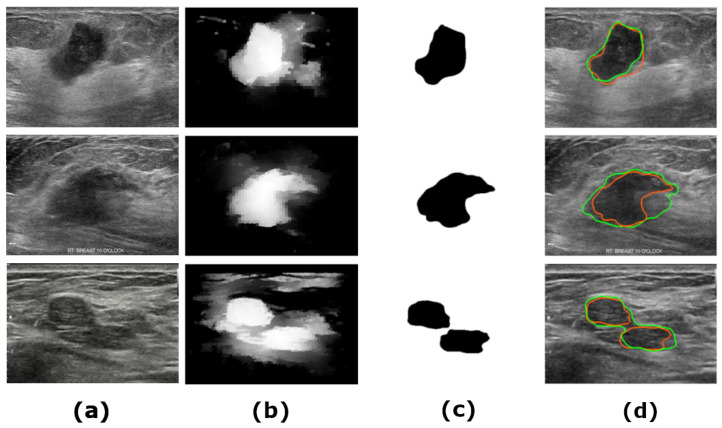
The attention mask: (**a**) a sample US image; (**b**) gray level attention map; (**c**) binarized attention mask; (**d**) red-mask, green-ground truth.

**Figure 2 diagnostics-13-03611-f002:**
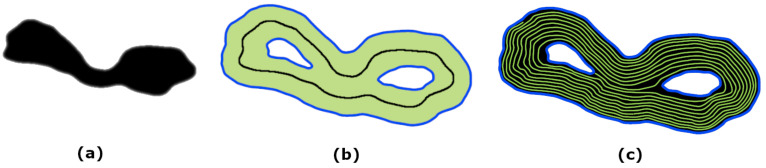
The offsets: (**a**) the mask, (**b**) the initial offset, (**c**) the generated offsets.

**Figure 3 diagnostics-13-03611-f003:**
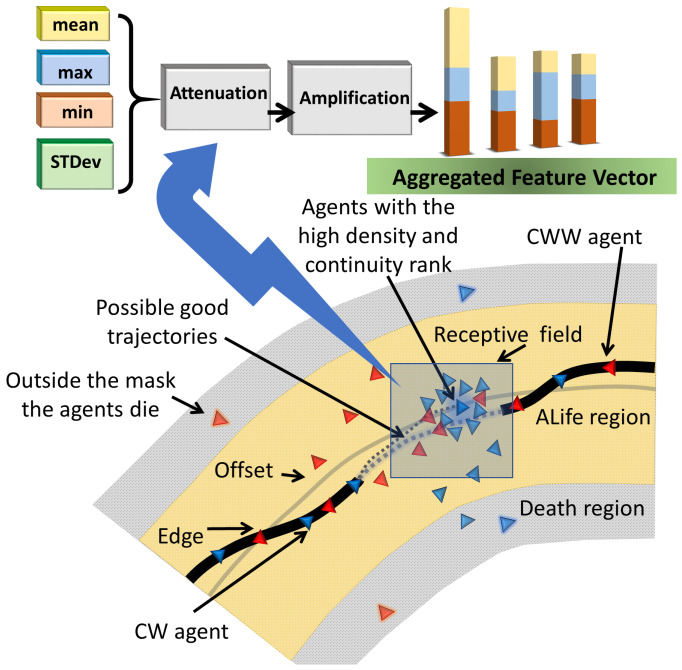
Key elements of the system.

**Figure 4 diagnostics-13-03611-f004:**
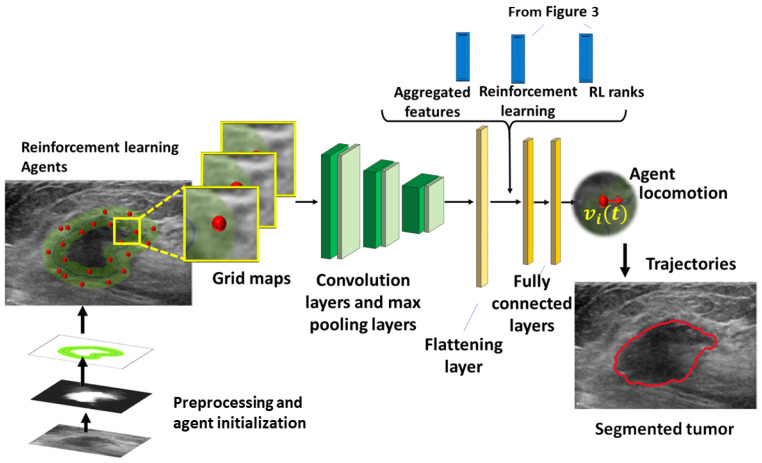
Backbone network of the model, red dots-agents, green AL region.

**Figure 5 diagnostics-13-03611-f005:**
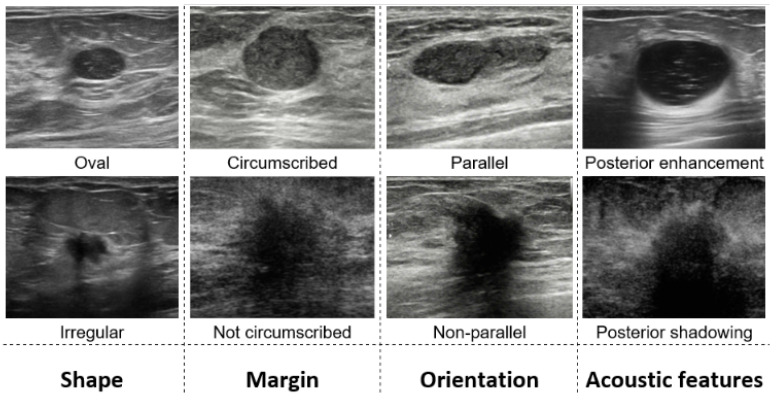
Conventional tumor categories.

**Figure 6 diagnostics-13-03611-f006:**
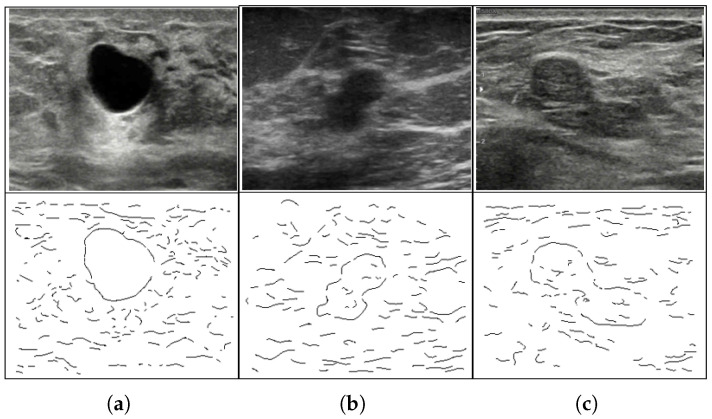
Raw US images and corresponding edge maps. (**a**) Simple cyst B|B|B, (**b**) medium complexity- B|T|T, (**c**) high complexity T|T|T.

**Figure 7 diagnostics-13-03611-f007:**
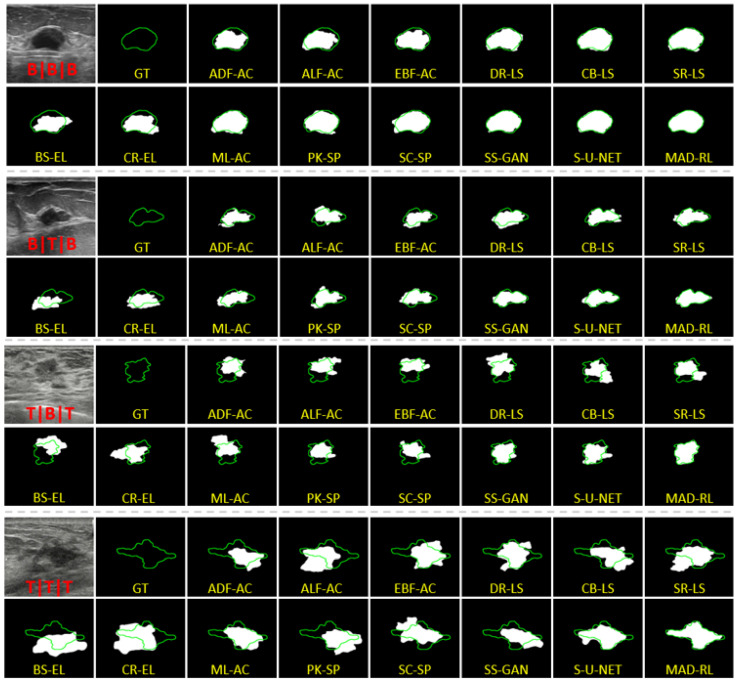
Segmentation of the tumors from different categories: white for ground truth, green for segmentation results.

**Figure 8 diagnostics-13-03611-f008:**
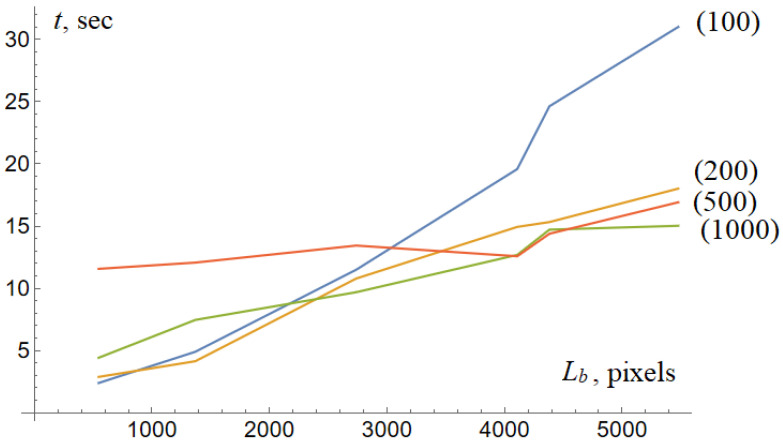
Computational time as a function of the length of the tumor for a variable number of the agents.

**Figure 9 diagnostics-13-03611-f009:**
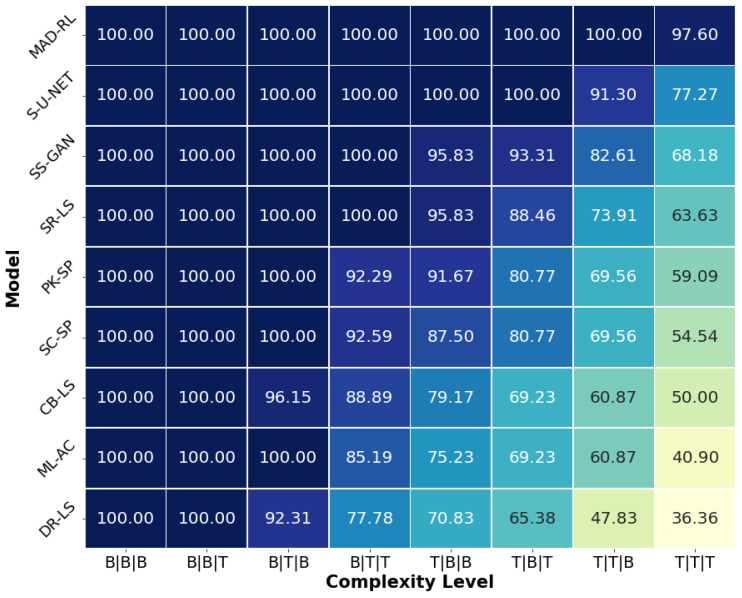
Heatmap of the segmentation ratio for different complexities. The best eight models.

**Figure 10 diagnostics-13-03611-f010:**
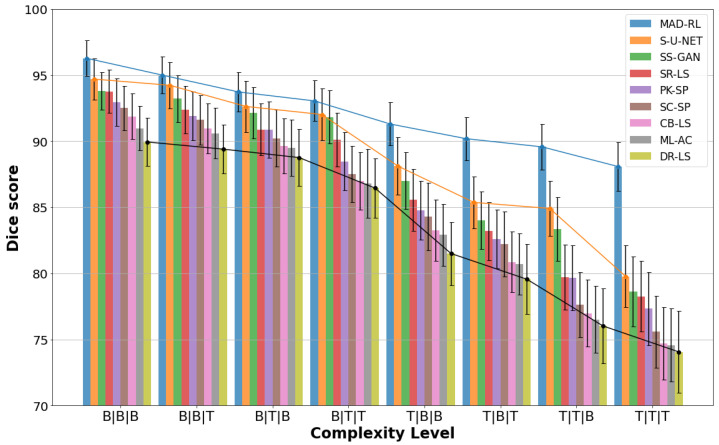
Dice index and standard deviations (whiskers) for different complexities.

**Table 1 diagnostics-13-03611-t001:** Hyperparameters of the training algorithm.

Parameter	Value	Description
$T_{max}$	900	max number of steps for collecting the data, Section 4
ϵ	0.2	clipping threshold, Section 4
γ	0.99	discount factor, Section 4
λ	0.95	smoothing factor, Section 4
$E_{\pi }$	40	number of epochs to update the policy, Section 4
$E_{\phi }$	40	number of epochs to update the value, Section 4
$lr_{\theta }$	$2e^{-4}$	learning rate of the policy network, Section 4
$lr_{\phi }$	$e^{-3}$	learning rate of the value network, Section 4
$t_{2}$	21	number of steps to evaluate the average curvature, Section 3.3
$\theta_{min}$	$118^{\circ }$	min collision angle, Section 3.4
*r*	24	size the receptive field (pixels), Section 3.3

**Table 2 diagnostics-13-03611-t002:** Multi-agent deep reinforcement learning vs. state-of-the-art segmentation, complexity level B|B|B.

Methods	${\mathrm{SG}}_{\mathrm{ratio}}$	$H_{3},\%$	Rec%	Pre%	Acc%	Jac%	Dice%
ADF-AC [12]	95.83	1.32 (±0.36)	95.76 (±2.35)	81.50 (±2.44)	97.54 (±0.62)	78.90 (±2.23)	88.18 (±1.94)
ALF-AC [14]	91.67	1.36 (±0.39)	95.04 (±2.38)	81.14 (±2.48)	97.54 (±0.64)	77.76 (±2.28)	87.48 (±1.89)
EBF-AC [13]	100.00	1.18 (±0.34)	95.91 (±2.31)	82.80 (±2.39)	97.82 (±0.61)	79.91 (±2.03)	88.82 (±1.73)
DR-LS [16]	100.00	1.11 (±0.31)	95.31 (±2.26)	84.24 (±2.34)	97.95 (±0.59)	80.84 (±2.12)	89.94 (±1.82)
CB-LS [19]	100.00	0.81 (±0.27)	97.71 (±1.94)	86.73 (±2.31)	98.43 (±0.63)	84.99 (±2.06)	91.88 (±1.71)
SR-LS [18]	100.00	0.56 (±0.22)	98.71 (±1.21)	89.30 (±2.09)	98.82 (±0.51)	88.28 (±1.92)	93.77 (±1.64)
BS-EL [20]	75.00	2.30 (±0.46)	91.65 (±2.82)	72.45 (±2.58)	96.09 (±0.68)	67.99 (±2.41)	80.91 (±2.02)
CR-EL [21]	83.33	1.79 (±0.42)	91.38 (±2.47)	77.51 (±2.53)	96.81 (±0.66)	72.19 (±2.37)	83.82 (±2.01)
ML-AC [35]	100.00	0.90 (±0.34)	95.89 (±2.22)	86.59 (±2.27)	98.27 (±0.57)	83.44 (±2.16)	90.97 (±1.68)
PK-SP [27]	100.00	0.66 (±0.28)	98.12 (±1.84)	88.33 (±2.14)	98.66 (±0.50)	86.84 (±1.98)	92.95 (±1.78)
SC-SP [3]	100.00	0.73 (±0.25)	97.64 (±1.63)	87.92 (±2.17)	98.57 (±0.52)	86.06 (±2.01)	92.51 (±1.67)
SS-GAN [36]	100.00	0.55 (±0.19)	97.88 (±1.78)	90.07 (±2.05)	98.83 (±0.46)	88.35 (±1.94)	93.80 (±1.42)
S-U-NET [37]	100.00	0.44 (±0.18)	98.84 (±1.12)	90.88 (±1.97)	99.00 (±0.44)	89.94 (±1.86)	94.70 (±1.55)
MADRL	100.00	0.27 (±0.08)	99.56 (±1.50)	92.40 (±1.43)	99.21 (±0.37)	92.81 (±1.40)	96.28 (±1.33)

**Table 3 diagnostics-13-03611-t003:** Multi-agent deep reinforcement learning vs. state-of-the-art segmentation, complexity level B|B|T.

Model	${\mathrm{SG}}_{\mathrm{ratio}}$	$H_{3},\%$	Rec%	Pre%	Acc%	Jac%	Dice%
ADF-AC [12]	91.67	1.34 (±0.38)	94.54 (±2.5)	81.79 (±2.59)	94.54 (±0.59)	77.86 (±2.33)	87.53 (±2.02)
ALF-AC [14]	87.50	1.44 (±0.39)	92.54 (±2.52)	82.14 (±2.63)	97.30 (±0.61)	76.71 (±2.31)	86.80 (±2.05)
EBF-AC [13]	91.67	1.19 (±0.36)	95.88 (±2.48)	83.11 (±2.55)	97.75 (±0.63)	80.37 (±2.25)	89.10 (±1.97)
DR-LS [16]	95.83	0.97 (±0.34)	95.90 (±2.45)	84.81 (±2.52)	97.97 (±0.55)	81.77 (±2.22)	89.4 (±1.86)
CB-LS [19]	100.00	0.80 (±0.31)	96.75 (±2.47)	85.90 (±2.43)	98.15 (±0.52)	83.44 (±2.15)	90.96 (±1.89)
SR-LS [18]	100.00	0.60 (±0.24)	96.88 (±2.28)	88.32 (±2.34)	98.48 (±0.47)	85.89 (±2.07)	92.39 (±1.81)
BS-EL [20]	66.67	2.46 (±0.47)	91.38 (±2.88)	71.99 (±2.69)	95.81 (±0.65)	67.44 (±2.47)	80.45 (±2.43)
CR-EL [21]	79.17	1.98 (±0.45)	90.81 (±2.55)	77.30 (±2.65)	96.52 (±0.68)	71.65 (±2.34)	83.42 (±2.28)
ML-AC [35]	100.00	0.83 (±0.38)	97.00 (±2.4)	85.02 (±2.45)	98.08 (±0.54)	82.83 (±2.18)	90.59 (±1.92)
PK-SP [27]	100.00	0.66 (±0.29)	96.73 (±2.31)	87.34 (±2.37)	98.36 (±0.48)	85.06 (±2.11)	91.92 (±1.85)
SC-SP [3]	100.00	0.72 (±0.25)	96.37 (±2.34)	87.46 (±2.40)	98.32 (±0.55)	84.55 (±2.13)	91.61 (±1.87)
SS-GAN [36]	100.00	0.52 (±0.25)	97.60 (±2.25)	89.23 (±2.32)	98.64 (±0.41)	87.29 (±2.14)	93.21 (±1.78)
S-U-NET [37]	100.00	0.43 (±0.21)	98.23 (±1.74)	90.19 (±1.95)	98.83 (±0.44)	89.09 (±1.98)	94.23 (±1.74)
MADRL	100.00	0.32 (±0.11)	99.02 (±1.56)	91.31 (±1.46)	99.01 (±0.38)	90.49 (±1.42)	95.00 (±1.39)

**Table 4 diagnostics-13-03611-t004:** Multi-agent deep reinforcement learning vs. state-of-the-art segmentation, complexity level B|T|B.

Model	${\mathrm{SG}}_{\mathrm{ratio}}$	$H_{3},\%$	Rec%	Pre%	Acc%	Jac%	Dice%
ADF-AC [12]	84.62	1.41 (±0.44)	93.65 (±2.71)	81.41 (±2.68)	97.35 (±0.57)	77.09 (±2.26)	87.05 (±2.24)
ALF-AC [14]	84.62	1.57 (±0.45)	90.54 (±2.75)	80.34 (±2.72)	96.98 (±0.59)	74.01 (±2.45)	85.04 (±2.56)
EBF-AC [13]	88.46	1.40 (±0.41)	95.66 (±2.67)	80.25 (±2.64)	97.34 (±0.56)	77.42 (±2.35)	87.25 (±2.22)
DR-LS [16]	92.31	1.13 (±0.39)	94.25 (±2.63)	83.97 (±2.60)	97.73 (±0.53)	79.80 (±2.27)	88.76 (±2.16)
CB-LS [19]	96.15	0.98 (±0.37)	95.85 (±2.55)	84.27 (±2.50)	97.88 (±0.50)	81.22 (±2.19)	89.62 (±2.08)
SR-LS [18]	100.00	0.82 (±0.33)	96.30 (±2.43)	86.10 (±2.38)	98.15 (±0.46)	83.28 (±1.93)	90.88 (±1.97)
BS-EL [20]	65.38	2.61 (±0.51)	90.78 (±2.88)	70.80 (±3.11)	95.52 (±0.63)	65.94 (±2.63)	79.46 (±2.84)
CR-EL [21]	73.08	2.25 (±0.46)	91.53 (±2.79)	73.44 (±2.76)	96.00 (±0.61)	68.70 (±2.59)	81.42 (±2.78)
ML-AC [35]	96.15	1.02 (±0.38)	96.61 (±2.59)	83.43 (±2.54)	97.84 (±0.51)	81.01 (±2.05)	89.50 (±2.12)
PK-SP [27]	100.00	0.84 (±0.34)	96.83 (±2.47)	85.64 (±2.42)	98.14 (±0.47)	83.30 (±1.93)	90.88 (±2.13)
SC-SP [3]	100.00	0.90 (±0.36)	95.22 (±2.51)	85.88 (±2.46)	98.04 (±0.49)	82.20 (±1.98)	90.22 (±2.14)
SS-GAN [36]	100.00	0.71 (±0.31)	97.48 (±2.40)	87.13 (±2.35)	98.38 (±0.45)	85.19 (±2.12)	92.15 (±1.94)
S-U-NET [37]	100.00	0.63 (±0.33)	97.57 (±2.36)	88.18 (±2.31)	98.51 (±0.44)	86.27 (±1.82)	92.63 (±1.95)
MADRL	100.00	0.43 (±0.15)	98.57 (±1.70)	89.35 (±1.60)	98.74 (±0.42)	88.19 (±1.44)	93.73 (±1.48)

**Table 5 diagnostics-13-03611-t005:** Multi-agent deep reinforcement learning vs. state-of-the-art segmentation, complexity level B|T|T.

Model	${\mathrm{SG}}_{\mathrm{ratio}}$	$H_{3},\%$	Rec%	Pre%	Acc%	Jac%	Dice%
ADF-AC [12]	70.37	1.70 (±0.45)	92.31 (±2.73)	79.41 (±2.71)	96.72 (±0.62)	74.41 (±2.38)	85.32 (±2.33)
ALF-AC [14]	66.67	1.87 (±0.47)	90.53 (±2.77)	78.89 (±2.75)	96.49 (±0.64)	72.72 (±2.52)	84.20 (±2.57)
EBF-AC [13]	77.78	1.69 (±0.43)	93.61 (±2.69)	78.52 (±2.67)	96.70 (±0.60)	74.47 (±2.44)	85.35 (±2.39)
DR-LS [16]	77.78	1.58 (±0.41)	93.19 (±2.65)	80.74 (±2.63)	96.99 (±0.58)	76.14 (±2.31)	86.45 (±2.25)
CB-LS [19]	88.89	1.44 (±0.37)	93.10 (±2.57)	81.73 (±2.55)	97.15 (±0.54)	76.99 (±2.22)	86.98 (±2.17)
SR-LS [18]	100.00	0.98 (±0.35)	95.66 (±2.45)	85.22 (±2.43)	97.84 (±0.48)	82.00 (±2.1)	90.11 (±2.05)
BS-EL [20]	55.56	2.79 (±0.51)	88.48 (±2.91)	70.32 (±2.83)	94.92 (±0.68)	64.16 (±2.79)	78.12 (±3.11)
CR-EL [21]	62.96	2.36 (±0.49)	88.32 (±2.81)	74.82 (±2.79)	95.70 (±0.66)	67.91 (±2.76)	80.87 (±3.01)
ML-AC [35]	85.19	1.46 (±0.39)	91.71 (±2.61)	82.53 (±2.59)	97.13 (±0.56)	76.68 (±2.46)	86.79 (±2.61)
PK-SP [27]	96.29	1.24 (±0.36)	94.92 (±2.49)	82.95 (±2.47)	97.45 (±0.50)	79.35 (±2.14)	88.48 (±2.21)
SC-SP [3]	92.59	1.43 (±0.38)	94.90 (±2.53)	81.28 (±2.51)	97.19 (±0.52)	77.78 (±2.18)	87.50 (±2.13)
SS-GAN [36]	100.00	0.77 (±0.33)	97.08 (±2.41)	87.15 (±2.39)	98.23 (±0.46)	84.91 (±2.06)	91.83 (±2.01)
S-U-NET [37]	100.00	0.74 (±0.32)	97.64 (±2.37)	87.03 (±2.35)	98.26 (±0.44)	85.23 (±2.02)	92.02 (±1.97)
MADRL	100.00	0.53 (±0.17)	97.17 (±1.72)	89.30 (±1.68)	98.51 (±0.49)	87.01 (±1.56)	93.05 (±1.54)

**Table 6 diagnostics-13-03611-t006:** Multi-agent deep reinforcement learning vs. state-of-the-art segmentation, complexity level T|B|B.

Model	${\mathrm{SG}}_{\mathrm{ratio}}$	$H_{3},\%$	Rec%	Pre%	Acc%	Jac%	Dice%
ADF-AC [12]	62.50	1.79 (±0.51)	88.30 (±2.79)	74.48 (±2.77)	95.76 (±0.67)	67.78 (±2.54)	80.77 (±2.47)
ALF-AC [14]	62.50	1.99 (±0.53)	83.72 (±2.83)	75.08 (±2.81)	95.53 (±0.69)	65.37 (±2.58)	79.01 (±2.51)
EBF-AC [13]	66.67	1.80 (±0.49)	87.91 (±2.75)	75.04 (±2.73)	95.80 (±0.65)	67.87 (±2.57)	80.84 (±2.65)
DR-LS [16]	70.83	1.61 (±0.47)	89.01 (±2.71)	75.26 (±2.69)	95.91 (±0.63)	68.79 (±2.46)	81.50 (±2.39)
CB-LS [19]	79.17	1.51 (±0.43)	93.18 (±2.63)	75.39 (±2.61)	96.22 (±0.59)	71.37 (±2.38)	83.27 (±2.31)
SR-LS [18]	95.83	1.15 (±0.37)	90.37 (±2.51)	81.31 (±2.49)	96.92 (±0.53)	74.81 (±2.26)	85.57 (±2.34)
BS-EL [20]	45.83	2.97 (±0.57)	79.37 (±2.91)	62.78 (±2.89)	93.47 (±0.73)	51.43 (±3.01)	67.89 (±3.23)
CR-EL [21]	54.17	2.45 (±0.55)	85.86 (±2.87)	69.86 (±2.85)	94.78 (±0.71)	62.32 (±2.98)	76.77 (±3.32)
ML-AC [35]	75.23	1.54 (±0.45)	92.58 (±2.67)	75.16 (±2.65)	96.15 (±0.61)	70.85 (±2.42)	82.91 (±2.35)
PK-SP [27]	91.67	1.34 (±0.39)	93.02 (±2.55)	78.03 (±2.53)	96.64 (±0.55)	73.63 (±2.3)	84.78 (±2.23)
SC-SP [3]	87.50	1.47 (±0.41)	91.20 (±2.59)	78.50 (±2.57)	96.58 (±0.57)	72.91 (±2.44)	84.30 (±2.57)
SS-GAN [36]	95.83	1.08 (±0.35)	93.39 (±2.47)	81.57 (±2.45)	97.19 (±0.51)	77.03 (±2.22)	87.01 (±2.15)
S-U-NET [37]	100.00	1.04 (±0.33)	94.05 (±2.43)	82.97 (±2.41)	97.45 (±0.49)	78.80 (±2.18)	88.12 (±2.19)
MADRL	100.00	0.61 (±0.21)	96.17 (±1.78)	86.93 (±1.78)	98.03 (±0.56)	84.01 (±1.64)	91.30 (±1.63)

**Table 7 diagnostics-13-03611-t007:** Multi-agent deep reinforcement learning vs. state-of-the-art segmentation, complexity level T|B|T.

Model	${\mathrm{SG}}_{\mathrm{ratio}}$	$H_{3},\%$	Rec%	Pre%	Acc%	Jac%	Dice%
ADF-AC [12]	57.69	1.89 (±0.57)	85.42 (±2.91)	73.02 (±2.87)	95.17 (±0.68)	64.76 (±2.66)	78.59 (±2.65)
ALF-AC [14]	53.85	2.01 (±0.58)	86.34 (±2.93)	71.08 (±2.89)	94.92 (±0.73)	63.72 (±2.68)	77.82 (±2.67)
EBF-AC [13]	61.54	1.88 (±0.56)	88.18 (±2.88)	71.74 (±2.84)	95.13 (±0.66)	65.29 (±2.64)	78.98 (±2.63)
DR-LS [16]	65.38	1.82 (±0.54)	89.81 (±2.86)	71.56 (±2.82)	95.21 (±0.64)	66.06 (±2.62)	79.55 (±2.65)
CB-LS [19]	69.23	1.69 (±0.44)	87.44 (±2.73)	75.31 (±2.46)	95.69 (±0.59)	67.89 (±2.29)	80.86 (±2.28)
SR-LS [18]	88.46	1.49 (±0.39)	89.16 (±2.65)	78.19 (±2.38)	96.27 (±0.53)	71.25 (±2.21)	83.19 (±2.19)
BS-EL [20]	38.46	3.15 (±0.61)	68.98 (±2.97)	63.40 (±2.94)	93.11 (±0.74)	47.83 (±3.12)	64.69 (±3.41)
CR-EL [21]	46.15	2.50 (±0.60)	76.13 (±2.95)	60.21 (±2.92)	92.51 (±0.72)	48.71 (±3.11)	65.48 (±3.52)
ML-AC [35]	69.23	1.74 (±0.45)	90.52 (±2.69)	72.95 (±2.49)	95.51 (±0.60)	67.67 (±2.32)	80.71 (±2.31)
PK-SP [27]	80.77	1.52 (±0.41)	92.36 (±2.58)	74.83 (±2.40)	95.97 (±0.55)	70.38 (±2.23)	82.61 (±2.22)
SC-SP [3]	80.77	1.60 (±0.42)	87.12 (±2.64)	78.10 (±2.43)	96.09 (±0.57)	69.84 (±2.26)	82.22 (±2.45)
SS-GAN [36]	93.31	1.47 (±0.37)	93.58 (±2.49)	76.32 (±2.35)	96.31 (±0.51)	72.44 (±2.18)	84.01 (±2.16)
S-U-NET [37]	100.00	1.24 (±0.36)	92.63 (±2.55)	79.30 (±2.05)	96.70 (±0.49)	74.49 (±2.14)	85.37 (±1.97)
MADRL	100.00	0.74 (±0.22)	95.53 (±1.77)	85.46 (±1.77)	97.87 (±0.58)	82.18 (±1.66)	90.19 (±1.62)

**Table 8 diagnostics-13-03611-t008:** Multi-agent deep reinforcement learning vs. state-of-the-art segmentation, complexity level T|T|B.

Model	${\mathrm{SG}}_{\mathrm{ratio}}$	$H_{3},\%$	Rec%	Pre%	Acc%	Jac%	Dice%
ADF-AC [12]	39.13	2.16 (±0.65)	76.67 (±2.96)	70.54 (±3.01)	92.69 (±0.75)	56.28 (±2.83)	71.94 (±2.93)
ALF-AC [14]	34.78	2.28 (±0.67)	76.67 (±2.98)	72.90 (±3.03)	92.89 (±0.77)	55.4 (±2.85)	71.31 (±2.92)
EBF-AC [13]	39.13	2.13 (±0.63)	82.28 (±2.93)	66.60 (±2.99)	91.92 (±0.73)	57.98 (±2.81)	73.36 (±2.88)
DR-LS [16]	47.83	1.97 (±0.61)	89.23 (±2.91)	66.36 (±2.97)	92.40 (±0.71)	61.39 (±2.79)	76.02 (±2.86)
CB-LS [19]	60.87	1.85 (±0.49)	88.33 (±2.84)	68.39 (±2.55)	92.82 (±0.69)	62.62 (±2.44)	76.98 (±2.54)
SR-LS [18]	73.91	1.59 (±0.43)	88.07 (±2.68)	73.21 (±2.49)	93.90 (±0.63)	66.28 (±2.38)	79.71 (±2.44)
BS-EL [20]	17.39	3.29 (±0.71)	75.39 (±3.02)	60.19 (±3.08)	90.63 (±0.81)	46.05 (±3.32)	63.06 (±3.78)
CR-EL [21]	26.09	2.73 (±0.69)	70.33 (±3.04)	67.24 (±3.06)	91.48 (±0.79)	51.6 (±3.29)	64.89 (±3.58)
ML-AC [35]	60.87	1.92 (±0.5)	83.56 (±2.78)	70.78 (±2.57)	93.11 (±0.7)	62.05 (±2.46)	76.51 (±2.52)
PK-SP [27]	69.56	1.76 (±0.45)	88.67 (±2.69)	72.56 (±2.51)	93.86 (±0.65)	66.25 (±2.4)	79.68 (±2.46)
SC-SP [3]	69.56	1.82 (±0.47)	85.38 (±2.72)	71.38 (±2.53)	93.39 (±0.67)	63.52 (±2.42)	77.63 (±2.48)
SS-GAN [36]	82.61	1.54 (±0.41)	90.35 (±2.71)	77.59 (±2.47)	95.13 (±0.61)	71.5 (±2.36)	83.36 (±2.42)
S-U-NET [37]	91.30	1.36 (±0.38)	93.22 (±2.66)	78.03 (±2.17)	95.51 (±0.58)	73.79 (±2.24)	84.91 (±2.09)
MADRL	100.00	0.84 (±0.25)	94.68 (±1.83)	85.08 (±1.86)	97.52 (±0.65)	81.15 (±1.75)	89.57 (±1.72)

**Table 9 diagnostics-13-03611-t009:** Reinforcement learning vs. state-of-the-art segmentation, complexity level T|T|T.

Model	${\mathrm{SG}}_{\mathrm{ratio}}$	$H_{3},\%$	Rec%	Pre%	Acc%	Jac%	Dice%
ADF-AC [12]	18.18	2.41 (±0.75)	78.41 (±3.09)	64.70 (±3.15)	91.22 (±0.84)	54.63 (±3.08)	70.57 (±3.15)
ALF-AC [14]	13.63	2.42 (±0.77)	84.84 (±3.11)	59.59 (±3.17)	90.40 (±0.86)	53.85 (±3.10)	70.00 (±3.17)
EBF-AC [13]	27.27	2.45 (±0.73)	82.88 (±3.07)	62.01 (±3.13)	90.76 (±0.82)	54.86 (±3.06)	70.82 (±3.13)
DR-LS [16]	36.36	2.32 (±0.71)	83.47 (±3.05)	67.21 (±3.11)	92.10 (±0.80)	58.83 (±3.04)	74.05 (±3.11)
CB-LS [19]	50.00	2.15 (±0.60)	83.02 (±2.89)	68.08 (±2.69)	92.37 (±0.78)	59.66 (±2.67)	74.71 (±2.74)
SR-LS [18]	63.63	1.92 (±0.54)	85.80 (±2.73)	72.23 (±2.63)	93.55 (±0.72)	64.31 (±2.61)	78.26 (±2.68)
BS-EL [20]	0.00	-	-	-	-	-	-
CR-EL [21]	0.00	-	-	-	-	-	-
ML-AC [35]	40.90	2.18 (±0.61)	82.74 (±2.81)	68.36 (±2.71)	92.37 (±0.79)	59.49 (±2.69)	74.57 (±2.76)
PK-SP [27]	59.09	2.05 (±0.56)	88.93 (±2.77)	68.68 (±2.65)	92.97 (±0.74)	63.11 (±2.63)	77.34 (±2.77)
SC-SP [3]	54.54	2.15 (±0.58)	87.90 (±2.86)	66.48 (±2.67)	92.30 (±0.76)	60.81 (±2.65)	75.58 (±2.72)
SS-GAN [36]	68.18	1.87 (±0.52)	88.44 (±2.78)	71.07 (±2.61)	93.52 (±0.71)	64.82 (±2.59)	78.61 (±2.66)
S-U-NET [37]	77.27	1.44 (±0.45)	89.68 (±2.68)	71.91 (±2.31)	93.81 (±0.67)	66.33 (±2.27)	79.76 (±2.34)
MADRL	97.60	0.90 (±0.30)	93.79 (±1.93)	83.27 (±1.92)	96.79 (±0.72)	78.94 (±1.85)	88.08 (±1.84)

**Table 10 diagnostics-13-03611-t010:** Reward weights.

Parameter	Value	Description
$w_{c,1}$	0.31	continuity by the curvature
$w_{c,2}$	0.23	continuity by collisions
$w_{pr}$	0.15	proximity
$w_{d}$	0.19	density
$w_{cl}$	0.12	closure

## Data Availability

The datasets were sourced from http://www.onlinemedicalimages.com (accessed on 17 June 2023) of Thammasat University Hospital, Thailand, Bangkok, and from https://www.ultrasoundcases.info (accessed on 20 July 2023) of The Gelderse Vallei Hospital, Ede, The Netherlands.

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
