# Peer review of "Edge-Driven Multi-Agent Reinforcement Learning: A Novel Approach to Ultrasound Breast Tumor Segmentation"

_diagnostics, 2023, doi:10.3390/diagnostics13243611_

Round 1

Reviewer 1 Report

Comments and Suggestions for Authors

The paper proposes a breast ultrasound image segmentation model using virtual agents (robots) trained via reinforcement learning. These agents, operating on the edge map, accurately delineate tumor contours and avoid false boundaries. The model outperforms 13 state-of-the-art models, effectively handling edge leaks and speckle noise typical in ultrasound images.

Some comments:

·        Some sentences are quite long and packed with information, which can make them hard to follow. Consider breaking them up or restructuring for clarity.

·        Early in the paper, perhaps in the introduction, outline the key contributions in bullet-point or numbered-list format. This gives the reader an immediate sense of the paper's significance.

·        The article lacks a sensitivity analysis for the reward function parameters. Different settings of these parameters could produce varying behaviors and performances for the agent. A comprehensive sensitivity analysis could offer insights into the robustness of the chosen reward function and its implications for the overall results.

·        The conclusion could be strengthened by providing a more integrative summary that ties together the key findings, implications, and contributions of the research.

Comments on the Quality of English Language

Minor editing of English language required

Author Response

The cover letter and the response are attached.

I wish to thank R1 for the useful comments.   

Reviewer 2 Report

Comments and Suggestions for Authors

The paper concerns a new method for ultrasound images analysis (i.e., segmentation). This method is based on agents and reinforcement learning and was applied to breast cancer images. The Authors compare the new method with 13 other, known from the literature, methods in an extensive computational experiment. The results showed that the new method outperforms the others. As the Authors mentioned, it can be due to the fact that the agents are able to detect a structure (context) of the analyzed image, which is a valuable feature.

In general, the paper is well written and interesting. However, there are a number of small language and other errors in the text. Hence, the whole paper should be carefully read and corrected.

Moreover, it would be interesting if the Authors wrote a bit more about the scalability of the proposed method. They mentioned that for large images the required computational resources may increase considerably. What is the rate of this increase, i.e., what is computational complexity of the proposed method? The second question which arises is: are there any features of the method which make it especially well suited for breast cancer image analysis, i.e., are there any characteristics of these images which are exploited by the method?

Summarizing, in my opinion the paper could be considered for a possible publication after minor revision.

Comments on the Quality of English Language

The are some small language errors in the text.

Author Response

The cover letter and the response are attached. 

I wish to thank R2 for the insightful comments.   

Reviewer 3 Report

Comments and Suggestions for Authors

The study presents a segmentation model for breast ultrasound images using reinforcement learning-trained virtual agents. These agents effectively identify tumor boundaries, even in the presence of false boundaries, by connecting fragmented edges. Although each agent works independently to maximize rewards, they communicate and collaborate to efficiently segment complex images, including those with edge leaks and speckle noise. The proposed model outperforms 13 state-of-the-art models, including deep learning variants and their modifications. There is a lot of valuable information in this study, and I think it is comprehensive. I find the subject fascinating. To improve the scientific level of the article, the following significant corrections seem necessary.

1- What is the motivation behind the work? This problem does not have a proper explanation. It is important that the authors demonstrate a scientific interest in the objectives and results of the study.

2. Provide more details on the proposed approach with a focus on the relationships between its components, since these are the key components of the solution and require more justification.

3- Write a better conclusion summarizing what has been learned and why it is valuable and exciting.

4- It would be helpful to revise the "Introduction" section to provide an accurate and informative literature review of the benefits and drawbacks of the existing methods and how the proposed method differs from them. Additionally, the motivation and contribution should be clarified.

5- The caption of the figures should be more informative.

6- To enhance the clarity of Figure 6 and emphasize the primary contribution and idea of the proposed network, I recommend making the main concept prominently visible within the network's structure. This will allow readers to readily identify and grasp the core idea of our approach.

Author Response

The cover letter and the response are attached.

I wish to thank R3 for the insightful comments. 

Round 2

Reviewer 3 Report

Comments and Suggestions for Authors

From my perspective, the authors addressed all comments.